# Cytomegalovirus Infections after Hematopoietic Stem Cell Transplantation: Current Status and Future Immunotherapy

**DOI:** 10.3390/ijms20112666

**Published:** 2019-05-30

**Authors:** Sung-Yeon Cho, Dong-Gun Lee, Hee-Je Kim

**Affiliations:** 1Division of Infectious Diseases, Department of Internal Medicine, College of Medicine, The Catholic University of Korea, Seoul 06591, Korea; cho.sy@catholic.ac.kr; 2Vaccine Bio Research Institute, College of Medicine, The Catholic University of Korea, Seoul 06591, Korea; 3Catholic Hematology Hospital, College of Medicine, The Catholic University of Korea, Seoul 06591, Korea; 4Division of Hematology, Department of Internal Medicine, College of Medicine, The Catholic University of Korea, Seoul 06591, Korea; 5Leukemia Research Institute, College of Medicine, The Catholic University of Korea, Seoul 06591, Korea

**Keywords:** antiviral drugs, cell therapy, cytomegalovirus, hematopoietic cell transplantation, T lymphocyte, vaccine

## Abstract

Cytomegalovirus (CMV) infection after hematopoietic stem cell transplantation (HSCT) is one of the critical infectious complications related to host immune recovery. The spectrum of CMV infection is quite extensive, from asymptomatic CMV reactivation presenting mainly as CMV DNAemia to fatal CMV diseases involving gut, liver, lungs, or brain. In addition to organ involvement, CMV reactivation can exert indirect effects such as immunosuppression or graft failure that may result in the development of concurrent infectious complications. Currently, preemptive therapy, which is based on PCR-based monitoring of CMV from blood, is a mainstay enabling improvement in CMV-related outcomes. During the past decades, new antiviral drugs, clinical trials for prophylaxis in high-risk groups, and vaccines for preventing CMV infection have been introduced. In addition, data for immunologic monitoring and adoptive immunotherapy have also been accumulated. Here, we review the current status and recent updates in this field, with future perspectives including immunotherapy in HSCT recipients.

## 1. Cytomegalovirus Infection after Hematopoietic Stem Cell Transplantation

Hematopoietic stem cell transplantation (HSCT) is an effective and intensive treatment option that aims to cure the following categories of diseases: (1) non-malignant diseases, such as aplastic anemia or chronic granulomatous diseases, which result from the functional failure of bone marrow or marrow-derived cells, and (2) hematologic malignancies, including acute or chronic leukemia, lymphoma, multiple myeloma, and myeloproliferative neoplasms, which are more common indications for HSCT [1]. In the latter, HSCT is intended to eliminate malignant cells by the myeloablative effect of cytotoxic therapy—comprising intensive chemotherapy and/or total body irradiation (TBI)—and induce a graft-versus-malignancy (GVM) effect by providing anti-neoplastic immune cells that express tumor-specific or -associated antigens [2,3]. In the past, survival rates following HSCT were low. However, since the introduction of the concept for human leukocyte antigen (HLA), the outcomes of allogeneic HSCT have improved with decrease of graft failure and/or graft-versus-host disease (GVHD). Currently, allogeneic HSCT has been performed with hematopoietic cells from various sources (bone marrow, peripheral blood, and cord blood), donors (sibling, unrelated, and haploidentical), and tailored conditioning regimens (myeloablative or reduced intensity/toxicity). Although therapeutic techniques such as immunosuppressant use as well as advanced graft manipulation methods have improved the transplant performance, infectious complications still affect the prognosis of HSCT recipients, since the extent of immunodeficiency and number of immunocompromised patients has increased [2,3,4].

Cytomegalovirus (CMV) infection after HSCT is one of the fatal infectious complications related to host immune recovery. The spectrum of CMV infection is quite extensive, from CMV reactivation without any organ involvement presenting mainly as asymptomatic viremia, DNAemia, or antigenemia to CMV end-organ diseases such as esophagitis, gastroenteritis, hepatitis, retinitis, pneumonia, and encephalitis. In addition to direct end-organ involvement, CMV reactivation can indirectly affect graft failure or immunosuppression which may result in concurrent bacterial and/or fungal infections [5].

CMV reactivation rate, after HSCT, has been reported to be 30–70%, and may be associated with a higher non-relapse mortality rate (relative risk (RR), 1.61 to 1.95) [6,7,8,9,10,11,12]. Mortality related to fatal CMV disease is still as high as 45–60% in HSCT recipients; CMV pneumonia and encephalitis are particularly fatal, despite aggressive treatment using antiviral agents and adjunctive therapies [13,14,15,16,17,18]. Currently, preemptive therapy based on active monitoring of CMV from the blood of HSCT recipients is a mainstay that reports improvement in CMV-related outcomes [19,20]. During the past decades, both real world practice and clinical trials for prophylaxis in high-risk groups, new antiviral drugs, and vaccines for CMV infection have been performed. In addition, data for immunologic monitoring and adoptive immunotherapy have also been accumulated. Here, we review the current therapies and recent updates in this regard, with future perspectives in HSCT recipients.

## 2. Current Therapeutic Approaches

Management of CMV is categorized into prophylaxis, preemptive, and targeted treatment. While there have been recent studies for universal CMV prophylaxis in HSCT recipients, the efficacy is still controversial; one retrospective study in haploidentical HSCT setting assessed the effectiveness of prophylaxis, but the prophylactic regimens could be partially active against CMV (i.e., ganciclovir 5 mg/kg i.v. every 12 h at D-2, and valganciclovir 500 mg once daily from D-1 to D+4, and then valacyclovir 1 g three times a day from D+5 to D+100 for intermediate-dose group (*n* = 45)) in the aspect of drugs, dosage, and durations of prophylaxis [21]. Another prospective study compared six months of valganciclovir (900 mg/day) with a placebo, thereby showing a reduced incidence of CMV DNAemia level (>1000 copies/mL) or a 5-fold increase over the baseline in the prophylaxis group. However, there was no significant benefit for the composite endpoint of death, CMV disease, or other severe invasive infections. In this study, although valganciclovir prophylaxis was found to be effective in reducing CMV reactivation, it was not superior to the preemptive strategy in terms of reducing death and CMV disease. In addition, more patients received growth factors in the prophylaxis group due to cytopenia than in the control group [22]. Therefore, unlike solid organ transplantation (SOT), most HSCT centers still introduce a preemptive strategy rather than routine universal prophylaxis in HSCT recipients, owing to both the advantages and adverse drug reactions related to antiviral agents, mainly including myelosuppression. Although the CMV-related outcome has been improved with preemptive therapy, CMV remains the major cause of transplantation-related non-relapse mortality [11]. 

Currently available standard anti-CMV agents include ganciclovir, valganciclovir, foscarnet, and cidofovir (Table 1) [23]. The first-line treatment of CMV infection after HSCT consists of ganciclovir at induction doses for at least two weeks, followed by maintenance doses until CMV viremia, DNAemia, or antigenemia become undetectable without any evidence of organ involvement [24]. Oral valganciclovir is a useful alternative if the patient’s oral intake is sufficient, with tolerable gastrointestinal symptoms. Clinicians have to monitor for any accompanying neutropenia or thrombocytopenia. In case of adverse events or intolerance due to the primary drugs, second- or third-line agents may be used with foscarnet or cidofovir. However, these agents can also be associated with toxicity, such as renal toxicity or electrolyte imbalance. Another important reason for changing the antiviral agent is CMV refractoriness, which is defined as documented failure to achieve > 1 log_10_ decrease in CMV DNA level in blood or plasma after two or more weeks of treatment with active antiviral agents against CMV [25]. If the cases are compatible with CMV refractoriness, a laboratory test for detecting resistant CMV should be performed. While genotyping takes a considerable number of days, laboratory testing for mutations in *UL97* and/or *UL54* is currently the most helpful method. Although the incidence of resistant CMV has been reported to vary from 0% to 7.9% in HSCT recipients (from HLA-matched donors), it rises up to 14.5% in high-risk patients from haploidentical donors [26,27,28,29,30]. When the resistance gets identified, it is necessary to change the antiviral agents into susceptible drugs and consider possible add-on adjunctive therapies (such as leflunomide). Leflunomide shows antiviral activity against several viruses via protein kinase inhibition. The activity of leflunomide has been demonstrated in vitro for BK virus and CMV, even in a resistant CMV strain. It should be used as an adjunctive therapy along with anti-CMV drugs. The outcome of the rescue therapy for refractory and/or multidrug-resistant severe CMV diseases was variable [25]; randomized controlled trials would be required to assess the efficacy and tolerance of leflunomide for this indication.

CMV organ involvement should be evaluated when the patient complains of symptoms such as nausea, vomiting, diarrhea, dry cough, dyspnea, headache, and altered mental status. An ophthalmologic examination is mandatory for the detection of CMV retinitis, even without symptoms. Frequently, those symptoms should be distinguished from those of GVHD or other complications related to HSCT. In practice, both CMV organ involvement and GVHD can be diagnosed simultaneously. Acute and chronic GVHD is a major risk factor in CMV infection, followed by CMV-negative donor and CMV-positive recipient serostatus, unrelated or mismatched donor, or the use of monoclonal antibodies (such as alemtuzumab) [31,32,33]. If diagnosis is delayed, response to the treatment may be poor. Among the CMV diseases, prognosis varies according to the involved organ; that is, CMV gastritis generally presents good prognosis, whereas CMV encephalitis or pneumonia is fatal despite prolonged antiviral treatment. Regarding adjunctive therapy for CMV pneumonia, recommendations include the addition of intravenous CMV or polyclonal immunoglobulin (Table 1) [20,34,35,36]. However, the use of immunoglobulin for other indications, except for CMV pneumonia, is still uncertain. In such complicated cases, treatment options are very limited and the experience of the clinician is vital.

## 3. New Antiviral Agents

Recently, clinical studies on novel antiviral agents for CMV have been established. Although the major strategy of CMV treatment is currently based on preemptive therapy, studies have been initiated based on whether the new antiviral agents are effective as a prophylaxis in HSCT recipients (Table 2).

Letermovir is a highly selective anti-CMV agent, which inhibits the cleavage of viral DNA and its packaging into capsids by targeting the CMV terminase complex [20,37]. The drug has an advantage of being administered either orally or intravenously, and can be taken once a day. It is active against both wild-type CMV and drug-resistant strains in vitro [20]. In phase 2 studies of letermovir prophylaxis in HSCT recipients, significant reduction in the incidence of CMV infection was seen in the letermovir group (60 mg, 120 mg, or 240 mg once daily; 46%, 32%, and 29% of all-cause prophylaxis failure, defined as drug discontinuation due to CMV infection, or disease, or any other cause) compared to that in a placebo (64%) [38]. In a phase 3 prophylaxis study, a total of 565 CMV seropositive patients received letermovir (480 mg once a day, or 240 mg once a day in patients taking cyclosporine) or a placebo for 14 weeks after HSCT. Letermovir prophylaxis resulted in a significantly lower risk of clinically relevant CMV infection (37% vs. 60%) than the placebo. Myelotoxic and nephrotoxic events were similar in both letermovir and placebo groups [39]. As a result, the U.S. Food and Drug Administration, as well as the European Medicines Agency and the Ministry of Food and Drug Safety of Korea, approved the use of letermovir for the prevention of CMV infection and diseases in adult CMV-seropositive allogeneic HSCT recipients. One of the implications of this phase 3 trial was that approximately 30% of high-risk patients were in both groups, and the preventive effect of letermovir was clearly confirmed in patients with a high risk of CMV disease. However, a cell-culture based study demonstrated CMV resistance against letermovir, presenting a *UL56* mutation, that was induced under a suboptimal concentration of the drug [40,41]. Letermovir resistance tended to be developed earlier than ganciclovir in in vitro, which implicates lower genetic barrier to resistance. In some subjects experienced breakthrough viremia in the phase 2 and phase 3 studies of letermovir prophylaxis, *UL56* mutation was also detected, while no mutation was detected in the placebo group [38,39,42]. Therefore, the monitoring of the treatment response of letermovir prophylaxis and the genotyping of *UL56* would be necessary in future practice [25]. It has not yet been determined whether letermovir resistance mutation is dose-dependent or not.

Maribavir is an orally available drug that binds to CMV protein kinase *UL97*, inhibiting viral DNA encapsidation and the nuclear egress of viral particles from infected cells. An in vitro study revealed more potent antiviral activity against CMV compared to that of ganciclovir, and it was also found to be active against ganciclovir-resistant CMV [43,44,45]. A phase 2 study demonstrated that the incidence of CMV infection based on plasma CMV DNA was lower in all three doses (100 mg twice daily, 400 mg once daily, or 400 mg twice daily; 7%, 11%, and 19% of CMV infection, respectively) of the maribavir group compared to that in the placebo (46%) [46]. In addition, there was no adverse effect on neutropenia or thrombocytopenia during maribavir prophylaxis. Thus, the results suggested maribavir prophylaxis effectively prevents CMV infection, leaving only a little concern regarding bone marrow suppression. However, in the phase 3 study, maribavir prophylaxis was no longer superior to the placebo [47]. There may be several possible causes for such a negative result in the maribavir prophylaxis study; inadequate dose of maribavir (the lowest dose being 100 mg twice daily), exclusion of the high-risk group, highly sensitive PCR, and low CMV disease rate in the control group may relate to active preemptive therapy [48]. More recently, clinical data on the use of maribavir for the treatment of refractory or resistant CMV have emerged, requiring further accumulation of data for transplantation recipients [37,49,50,51].

Brincidofovir is a lipid-conjugated analog of cidofovir that has high oral bioavailability and long half-life. The characteristics of brincidofovir include a dosage of twice a week and more efficient penetration into cerebrospinal fluid than ganciclovir or foscarnet due to the lipid conjugation. Moreover, it is known to show lower renal toxicity than cidofovir and broad spectrum against herpesviruses, polyomaviruses, adenoviruses, papillomaviruses, and variola virus [49,52,53]. A phase 2 study of brincidofovir (CMX001) in HSCT recipients showed a significantly lower incidence of CMV events in the brincidofovir group (100 mg twice weekly; 10% of patients developed CMV infection or disease) than in the placebo group (37%). Neither myelosuppression nor nephrotoxicity was observed in this study, and diarrhea was dose-limiting at a dose of 200 mg twice weekly [54]. In a phase 3 study of oral brincidofovir (100 mg twice weekly) for 14 weeks post HSCT, no clinically significant reduction of CMV reactivation and disease was seen over 24 weeks (51.2% and 52.3% in the brincidofovir and placebo groups, respectively). However, there were fewer detectable CMV reactivations (30% and 42.3%) and less need for preemptive therapy (29% and 37.6%) [55]. 

## 4. Cytomegalovirus and Host Immune Response

CMV is a ubiquitous β-herpesvirus that replicates only in human cells. Like other herpesviruses, it is characterized by latency after primary infection, which can be reactivated particularly in the setting of immune suppression [56]. The primary infection of CMV activates the innate immune response, which causes the release of inflammatory cytokines and co-stimulatory molecules from monocytes, macrophages, and dendritic cells (DCs). These processes work to slow down viral replication before the adaptive immune response develops adequately. Once the virus disseminates to cells of myeloid lineage, including monocytes and CD34 cells, it establishes a latent infection [57]. CMV-specific CD4 T cells have been described as appearing one week after the peak of CMV replication, and synthesize T helper 1 type (Th1) cytokines including interferon-γ (IFN-γ) and tumor necrosis factor-α (TNF-α). Then, CMV-specific CD8 T cells become detected in peripheral blood. These virus-specific CD8 T cells have the capacity of lysing CMV peptide-presenting target cells. In the months following primary infection, CMV-specific CD8 T cells acquire effector memory T cells. Interestingly, these T cells do not eliminate the virus even if latent infection persists in the host (Figure 1).

As described above, in most hematologic diseases following HSCT, immune cells of the recipient are eliminated along with the malignant or defective cells during the conditioning process. Through engraftment of donor stem cells, restoration of adaptive immunity proceeds slowly over a period of months to 1–2 years. CMV seropositive allogeneic HSCT recipients are the highest risk group of CMV infection, followed by SOT recipients, patients with active HIV, and patients that received T-cell depletion therapy including alemtuzumab, antithymocyte globulin (ATG), or post-transplant cyclophosphamide [33]. The immune system of the recipient changes with time after allogeneic HSCT. The transferred immunity from the graft is maintained for a limited period of time during the early phase post-HSCT, followed by a gradual decrease and immune reconstitution continues for a period of months to years. Therefore, the immunologic status of the donor is very important in terms of transferred immunity during the early phase after transplantation [58].

The decreased immune function of the recipient is caused by the underlying hematologic disease itself, drugs administered during the conditioning process (i.e., ATG), immunosuppressive agents after transplantation, and acute or chronic GVHD. After transplantation, the immune system begins to recover with neutrophil engraftment, but the functional recovery of lymphocytes occurs over a long period of time. The following examples show how the immunologic status of an HSCT recipient varies across cases. In some patients, CMV viremia or DNAemia may be smoothly improved by short-term preemptive antiviral therapy or by the reduction of immunosuppressants. However, in other patients, viral reactivation may persist for more than several weeks or progress to fatal CMV diseases, despite multi-directional approach, including adequate antiviral therapy with a reduction of immunosuppressants. Considering these differences, CMV-specific cell-mediated immunity (CMI) exerts an important role in the relationship between CMV and the host.

Although it is difficult to measure the accurate extent of immune reconstitution after transplantation, attempts have been made to measure CMV-specific CMI in SOT or HSCT recipients. Studies addressing CMV-specific immune recovery after transplantation have been performed based on the tetramer assay, intracellular cytokine staining (ICC), enzyme-linked immunospot (ELISpot), lymphocyte proliferation assay, QuantiFERON, etc. [59]. Clinicians may obtain different results depending on the method selected to measure CMV-specific CMI; tetramer assay is HLA-restricted, and multicolor flow cytometry after ICC can analyze not only the number of CMV-specific T cells but also various functional potentials; however, no standardization has been done yet. In particular, lymphopenia is common in HSCT recipients, and may affect the test results, thus posing as a general limitation in clinical practice.

While large-scale prospective studies are still lacking, CMV-specific CMI clearly seems to affect the development and prognosis of CMV infections. Cellular immunity driven by T cells is known to be responsible for controlling CMV replication, whereas a lack or delayed recovery of CMV-specific CD4- and CD8-T lymphocytes causes a predisposal to CMV recurrence and CMV disease [59,60,61,62,63,64,65,66]. CMV reactivation is usually related to a high frequency of GVHD, which might partially result in enhanced T cell reconstitution in patients with CMV infection. A recent study suggested that the presence of CMV-specific cytotoxic T lymphocytes (CTLs) in CMV seropositive recipients is associated with faster T cell reconstitution, which might enhance donor alloreactivity, thus contributing significantly to the elimination of residual host hematopoiesis reflected by complete donor chimerism [67]. Interestingly, there are reports that CMV reactivation is associated with protection from leukemic relapse, especially in acute myeloid leukemia [68,69,70,71]. This anti-leukemic effect might be mediated by CMV-driven expansion of donor-derived memory-like NKG2C+ NK and γδT cells [68].

In addition, the close association of CMV reactivation, GVHD, and donor chimerism can also be explained with the suppressing of cytokine signaling genes (*SOCS*), which is responsible for negative feedback regulators of cytokine signaling such as IFN-γ and interleukins and for defecting T cell homeostasis [72,73]. Of the eight SOCS proteins, SOCS1–SOCS7, cytokine-induced SRC-homology 2 (SH2) protein (CIS), SOCS1, and SOCS3 in toll-like receptor immune responses have been relatively well investigated. *SOCS1* is known to reduce the development of GVHD by inhibiting cytokine storm, as well as sustained engraftment of normal hematopoiesis. A recent study reported that the expression levels of *SOCS1* decreased in recipients with significant GVHD when compared to non-GVHD recipients. Also, the expressions of *SOCS1* decreased significantly more in chronic severe GVHD than in acute GVHD patients. In contrast, *SOCS3* expressions were similarly reduced in all the HSCT recipients [74]. Another study for the expression of *SOCS* genes showed higher *SOCS1* expression in CMV reactivated patients than other allogeneic HSCT recipients [75]. When the subgroup was divided according to the CMV and GVHD status, increased expression of *SOCS1* is exaggerated in the absence of GVHD (CMV+/GVHD-), whereas when GVHD is accompanied by CMV reactivation (CMV+/GVHD+), the increase is reduced (Figure 2). This result provides a new platform to study GVHD immunobiology and potential diagnostic and therapeutic targets for GVHD. In order to better understand the CMV-related outcome, along with the prognosis associated with allograft- and transplant-related outcomes, further studies are recommended for immune reconstitution and virus-host interactions.

Immune monitoring can be required for patients at high-risk of CMV disease, such as in haploidentical or double cord blood HSCT. In clinical practice, measurement of CMV-specific T-cell immunity after HSCT can be used to identify patients at risk of CMV-related complications [76,77]. Using CMV-specific CTL as a biomarker may facilitate decision-making based on risk assessment, and finally improve the outcome; however, further experience is warranted.

## 5. Vaccine Trials and Immunogenicity

The development of an effective and safe vaccine against CMV remains an important medical priority. Several studies are currently ongoing to develop an effective CMV vaccine [78,79,80,81,82]. Glycoprotein B (gB) of CMV is the main target of neutralizing antibodies [58]. The types of vaccines may include adjuvant recombinant protein vaccine based on envelope glycoprotein using DNA plasmid, or peptide-based vaccine, vectored vaccine, and peptide vaccine [79]; ASP0113, the DNA vaccine, is the most studied in HSCT recipients (Table 2).

ASP0113 is a first-in-class, bivalent DNA vaccine, designed for the prevention of CMV. It contains two plasmids, VCL-6365 and VCL-6368, encoding human CMV gB and phosphoprotein 65 (pp65) respectively, and formulated with CRL1005 poloxamer and benzalkonium chloride (BAK) [79,80,81]. A phase 2 study has demonstrated that ASP0113 significantly reduces the occurrence and recurrence of CMV viremia and improves the time to viremia episode compared to the placebo. Although the CMV-free rate at 1 year was higher in the vaccine group, the primary endpoint of clinically significant viremia resulting in the initiation of CMV-specific antiviral therapy was not achieved. According to the immunogenicity data, the number of pp65 interferon-γ-producing T cells was increased at all time-points after HSCT, although not statistically significant by repeated ANOVA measurements. gB IFN-γ-producing T cells were not significantly elicited in the vaccine group compared to that in the placebo group at all time-points after transplantation [80]. Currently, the results from the ASP0113 phase 3 study is being summarized, and subjects will continue to be followed up for 5.5 years post-transplantation.

Another vaccine, derived from the soluble recombinant gB with the adjuvant MF59, reduced the duration of viremia as well as days of ganciclovir treatment in liver or kidney transplantation recipients, only in CMV seronegative recipients of organs from seropositive donors [58,82]. There was a significant increase in gB antibodies after vaccination in both seronegative and seropositive recipients. This suggests that humoral immunity might play an important role in the seronegative-recipient subgroup; further studies would be required to elucidate the role of vaccine elicited antibody-dependent cell-mediated cytotoxicity (ADCC) for preventing the early and late reactivation of CMV.

Another trial was performed for the development of CMV monoclonal antibodies. CSJ148 is a mixture of two anti-CMV human monoclonal antibodies (LJP538 and LJP539) that bind to and inhibit the function of CMV gB and pentameric complex, consisting of glycoproteins gH, gL, UL128, UL130, and UL131. In a phase 1 clinical trial, CSJ148 was safe and well tolerated, with pharmacokinetics as expected for human immunoglobulin [83]. In the phase 2 study, the efficacy and safety of CSJ148 were evaluated for prophylaxis against HCMV in allogeneic HSCT recipients. Results of this phase 2 study are currently under review, and further development is now suspended (Table 2) [79].

## 6. Immunotherapy for Cytomegalovirus

Immunotherapy is a field of interest in various therapeutic areas in recent years. Although this is mainly developed for cancer treatment, attempts have been also made to control infectious diseases since post-HSCT infectious complications are also associated with immune deprivation before immunological reconstitution or persistent immune dysfunction. Among infectious diseases (i.e., bacterial, viral, fungal, etc.), immunotherapy has been advanced more particularly for viruses with latency and reactivating characteristics such as CMV and Epstein-Barr virus (EBV) [84]. Data on the number and functional reconstitution of virus-specific T cells, after allogeneic HSCT, have been accumulated over the past decades. The reconstitution of an antiviral T cell response can prevent CMV reactivation or diseases. The restoration of antiviral immunity by adoptive transfer of T cell clones, isolated from hematopoietic stem cell donors, has been attempted since the 1990s to prevent CMV viremia [85,86].

Techniques for the isolation of virus-specific T cells have been improved during the recent two decades. Traditionally, donor lymphocyte infusion (DLI), which had been used to secure anti-tumor immunity, can exert antiviral effects as well as induce significant GVHD due to alloreactive T cells [87]. This disadvantage for the risk of GVHD has led to the development of strategies to isolate virus-specific T cells. Virus-specific T cells can be generated by ex vivo culture from donor peripheral blood mononuclear cells (PBMC) [88]. To generate CMV-specific CD4 and CD8 T cells, in vitro stimulation and an expansion process are needed. The viral stimuli can be made via viral peptide, protein, lysate, or antigen presenting cells. In detail, there are several protocols using the co-culture of PBMC with CMV-infected fibroblast, pulsed autologous dendritic cells with viral lysate, and irradiated autologous virus-transformed B cell lines [88,89,90,91]. Then, virus-specific T cells can be expanded in vitro or direct infusion into the recipient and are supposed to be proliferated under physiological environment in vivo [88]. A method for the direct selection of a large amount of virus-specific T cells can also be used and categorized into two types. One is cytokine capture assay and the other is multimer (i.e., tetramer, pentamer, or streptamer) isolation method [92]. Cytokine capture assay yields a relatively low purity of CMV-specific T cells, and leads to increased concern of GVHD. However, correlation between the development of significant GVHD and purity was not established by the data [93,94]. In recent years, the use of automated devices such as CliniMACS^®^ system (Miltenyi Biotec, Germany) in the cytokine capture assay have enabled the simple and robust handling of virus-specific T cells [95]. The other method uses major histocompatibility complex (MHC) multimers, which show high purity, but cannot be used in rare HLA types [93].

Adoptive CMV-directed T cell therapy can target both prophylactic and therapeutic applications. A recent prospective multicenter study evaluated the safety and efficacy of CMV-specific CTLs, derived from a stem cell donor or third party donor, for the treatment of persistent CMV infection after allogeneic HSCT [96]. In T cell-depleted CMV seropositive donors, the results revealed the successful detection of CMV-specific CTLs and a complete or partial virological response post adoptive T cell therapy. In the case of T cell-replete CMV seropositive donors, the priority of the adoptive transfer of CTLs is low. If CMV-specific CTL therapy is performed from a third party in CMV seronegative stem cell donors, the rejection of incompletely HLA-matched T cells should be considered. Therefore, a high concordance of HLA molecules between the third party donor-derived cells and stem cell donor/recipient seems to be an important prerequisite for systemic survival of adoptively transferred T cells. Although cell therapy cannot be recommended as a standard at present, experimental adoptive immunotherapy, with donor or third party anti-CMV-specific T cells, may be available within the range of clinical trials. While there is no randomized trial and definite indication of immunotherapy, adoptive transfer of CMV-specific CTLs may be a valid therapeutic option if there is persistent or multidrug-resistant CMV infection with a suspected low level of CMV-specific CTLs. However, the efficacy in patients receiving high-dose corticosteroids is expected to be low. In addition, it is difficult to prepare the CTLs in a timely manner. The process needs to be established and improved for the future treatment. The possible indications of immunotherapy using CMV-specific CTLs might include refractory CMV infection in high-risk transplantation settings, such as cord blood transplantation or haploidentical HSCT [97]. Ongoing clinical trials for CMV-specific CTLs or multi-virus targeting CTLs will give us further advances in managing difficult-to-treat CMV infection in the near future.

## 7. Conclusions

During recent decades, there has been much advancement in the management of CMV infection, including the development of new drugs, prophylaxis using novel anti-CMV drugs and vaccines, and adoptive transfers of CMV-specific CTLs with immunologic monitoring. Although newer agents are not necessarily superior to the current standard of care, it is important to identify challenging cases in high-risk patients and find opportunities to improve the outcome through the multidirectional treatment approaches discussed here. Further studies aiming for the detailed understanding of virus-host immune interaction will play a vital role in improving the outcome of CMV diseases. 

## Figures and Tables

**Figure 1 ijms-20-02666-f001:**
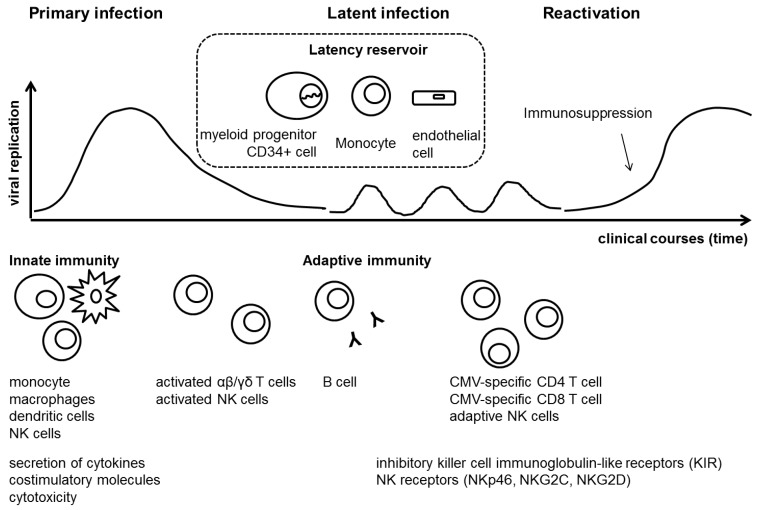
Overview of host immune response after cytomegalovirus infection.

**Figure 2 ijms-20-02666-f002:**
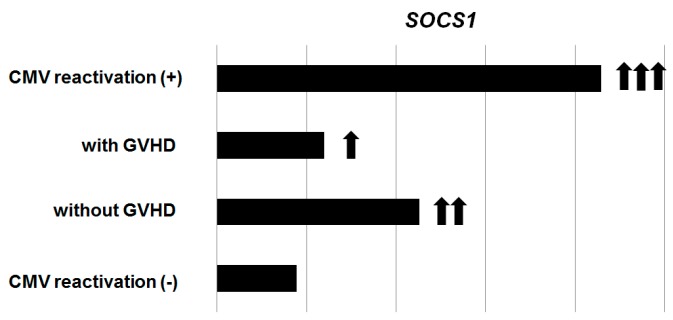
Expression of the *SOCS1* gene in allogeneic hematopoietic stem cell transplantation recipients. Different expression of *SOCS1* according to the cytomegalovirus reactivation and graft-versus-host diseases status (data modified from *Blood Res* 2015, 50, 40–45 [75]). Abbreviations: CMV, cytomegalovirus; GVHD, graft-versus-host diseases.

**Table 1 ijms-20-02666-t001:** Standard therapies against CMV.

Drugs	Mechanism	Dosing Regimens	Main Adverse Events and Considerations
Ganciclovir	Inhibits DNA polymerase (encoded by *UL54* gene, Needs to be phosphorylated by viral phosphotransferase (encoded by *UL97* gene)	Induction: 5 mg/kg IV every 12 h for at least 7–14 days Maintenance: 5 mg/kg IV once daily until test is negative Note: Minimum total induction and maintenance treatment is 2 weeks when 14 days of induction is used, and 3 weeks when a 7-day induction course is used.	Myelosuppression, Nephrotoxicity
Valganciclovir	Inhibits DNA polymerase, orally bioavailable formulation prodrug of ganciclovir	(Persons ≥40 kg with good oral intake) Induction: 900 mg PO twice daily for at least 14 days Maintenance: 900 mg PO once daily for 1–2 weeks until test is negative Note: Minimum treatment course is 14 days regardless of drug used	Myelosuppression, Nephrotoxicity
Foscarnet	Inhibits DNA polymerase *UL54* directly by this pyrophosphate analogue	Induction: 60 mg/kg IV every 8 h or 90 mg/kg every 12 h for 2–3 weeks Maintenance: 90 to 120 mg/kg once daily	Nephrotoxicity, Electrolyte imbalance, Myelosuppression
Cidofovir	Nucleotide analogue that inhibits DNA polymerase *UL54*	Induction: 5 mg/kg IV every weekly for 2 weeks Maintenance: 5 mg/kg IV every 2 weeks	Nephrotoxicity, Myelosuppression; Hydration and probenecid required to reduce nephrotoxicity
Leflunomide	Inhibits virion assembly, frequently used as add-on therapy	Loading dose: 100 mg orally once daily for 3 days only for patients at low risk for hepatotoxicity or myelosuppression Maintenance dose: 20 mg orally once daily; may reduce dose to 10 mg orally once daily if higher dose not tolerated	Liver cytolysis, Myelosuppression
CMV Ig/Polyclonal Ig	Increase CMV antibody levels	Varies among different studies and disease status	Infusion reactions

Abbreviation: CMV, cytomegalovirus; Ig, immunoglobulin.

**Table 2 ijms-20-02666-t002:** New antiviral agents or strategies for managing CMV.

Drugs	Mechanisms	Indication or Primary Endpoint in Clinical Trials, Dosing Regimens if Possible	Main Adverse Events and Considerations
Letermovir	CMV terminase inhibitor that targets the UL56 viral subunit	Prophylaxis of CMV infection and disease in CMV-seropositive recipients of an allogeneic HSCT 480 mg PO or IV once daily through 100 days post transplantation (coadministration with cyclosporine: If cyclosporine is initiated after starting letermovir, decrease the next letermovir dose to 240 mg once daily; if cyclosporine is discontinued after starting letermovir, increase the next letermovir dose to 480 mg once daily; if cyclosporine is interrupted due to high cyclosporine levels, no dose adjustment of letermovir is needed)	Nausea, vomiting Has not been studied as an agent for treatment; has multiple drug interactions; lacks activity against other herpesviruses including HSV and VZV
Maribavir	*UL97* viral protein kinase inhibitor	Not approved yet Clinical trials for treatment in transplant recipients with CMV infections that are refractory or resistant to treatment with ganciclovir, valganciclovir, foscarnet, or cidofovir; preemptive treatment in adult transplanted patients presenting with asymptomatic viremia	Taste disturbance Lower risk of hematotoxicity and absence of clear nephrotoxicity
Brincidofovir	Inhibits DNA polymerase, orally bioavailable formuation prodrug of cidofovir	Not approved yet Clinical trials for prophylactic or preemptive treatment of CMV infections; can be administered twice a week due to a long half-life	Diarrhea No excessive risk of nephro- and hematotoxocity
CMV vaccine	Stimulate CMV specific T cell immunity	Not approved yet Clinical trials for prophylaxis of CMV infection or reactivation in CMV seropositive patients undergoing allogeneic HSCT (ASP0113)	Minimal differences between the vaccine and placebo groups (ASP0113)
Passive CMV immune therapy	Monoclonal antibodies that block gB and others	Not approved yet Clinical trials for preventing of CMV infections in patients undergoing allogeneic HSCT recipients (CSJ148)	Nausea, Diarrhea, Vomiting, Stomatitis, and pyrexia (CSJ148)
Cell therapy	Adoptive transfer of CMV specific cytotoxic T lymphocytes	Not approved yet Clinical trials for treatment of persistent or refractory CMV infection after allogeneic HSCT	-

Abbreviations: CMV, cytomegalovirus; HSCT, hematopoietic stem cell transplantation; HSV, herpes simplex virus; VZV, varicella zoster virus; gB, glycoprotein B.

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
