# Peer review of "Cytomegalovirus Infections after Hematopoietic Stem Cell Transplantation: Current Status and Future Immunotherapy"

_ijms, 2019, doi:10.3390/ijms20112666_

Reviewer 1 Report

Need to improve quality of Fig 1.

In general , this article is well structured and developed.

All necessary items are precisely described .

Nevertheless , a certain inbalance on deepness and extension could be observed on item 6 which is, apparently, the main target of this paper. 

I'm recomending to improve this chapter.

Author Response

According to the reviewer's comment, we have revised the contents for immunotherapy (section 6 of the manuscript) with additional references.

The corrections include detailed strategies of adoptive T cell transfer to treat viral infection after HSCT with the history of the cell therapy (Line 326-349, Line 357-358, Line 367-377).  

The corrections are highlighted in the attatched file.

In addition, we have corrected the figure 1 as an additional figure file to improve the quality.

We hope the revised manuscript will better meet the requirement of the journal for publication.

We thank the editor and the reviewer of the IJMS once again.

Reviewer 2 Report

The review by Cho et al. nicely describe  the role of CMV infection/reactivation in HSCT and recent developments in new antiviral drugs and vaccines. However Major points are missing or not address adequate.

1.     The introduction part 1.1 is not necessary, because the main topic s CMV. On the other hand the part about CMV (1.2) is quite short. I would recommend to change the folding of the paras 3 and 4.

2.     Recently described resistances against leztermivir are not mentioned.

3.     Cellular based therapies to treat CMV-infections such as CD45RA-depletion as well as generation of T-cell lines or clones via expansion protocols. References mentioned here are not actual

Author Response

According to the reviewer's comment, we have revised the manuscript.

The changes are summerized below.

We have revised the introduction part with merging 1.1 and 1.2 into one section. (Line 35-71)

We have changed the folding of the paragraph 3 and paragraph 4 according to the reviewer's suggestion.

We have added descriptions for letermovir resistance. (Line 155-162)

We have modified the section 6 for immunotherapy including the contents for generation of T cell lines or clones via expansion protocols. (Line 326-349, Line 357-258, Line 367-377)

Round  2

Reviewer 1 Report

Modifications have clearly improved the manuscript.